# Molecular and Morphological Changes Induced by Leonardite-based Biostimulant in *Beta vulgaris* L.

**DOI:** 10.3390/plants8060181

**Published:** 2019-06-18

**Authors:** Valeria Barone, Giovanni Bertoldo, Francesco Magro, Chiara Broccanello, Ivana Puglisi, Andrea Baglieri, Massimo Cagnin, Giuseppe Concheri, Andrea Squartini, Diego Pizzeghello, Serenella Nardi, Piergiorgio Stevanato

**Affiliations:** 1Department of Agronomy, Food, Natural Resources, Animals and Environment, University of Padova, Viale Università, 16, 35020 Legnaro (PD), Italy; valeriabarone89@alice.it (V.B.); giovanni.bertoldo.learning@gmail.com (G.B.); chiarabr87@yahoo.it (C.B.); massimo.cagnin@unipd.it (M.C.); giuseppe.concheri@unipd.it (G.C.); squart@unipd.it (A.S.); diego.pizzeghello@unipd.it (D.P.); serenella.nardi@unipd.it (S.N.); 2SIPCAM S.p.A., S.S. Sempione, 195, 20016 Pero (MI), Italy; fmagro@sipcam.com; 3Department of Agriculture, Food and Environment, University of Catania, Via S. Sofia 98, 95123 Catania, Italy; ipuglisi@unict.it (I.P.); abaglie@unict.it (A.B.)

**Keywords:** sugar beet, hormonal metabolism, genetic expression, root growth

## Abstract

Humic substances extracted from leonardite are widely considered to be bioactive compounds, influencing the whole-plant physiology and the crop yield. The aim of this work was to evaluate the effect of a new formulate based on leonardite in the early stage of growth of sugar beet (*Beta vulgaris* L.). A commercial preparation of leonardite (BLACKJAK) was characterized by ionomic analysis, solid-state ^13^C MAS NMR spectroscopy. Seedlings of sugar beet were grown in Hoagland’s solution under controlled conditions. After five days of growth, an aliquot of the concentrated BLACKJAK was added to the solution to obtain a final dilution of 1:1000 (0.5 mg C L^−1^). The sugar beet response in the early stage of growth was determined by evaluating root morphological traits as well as the changes in the expression of 53 genes related to key morphophysiological processes. Root morphological traits, such as total root length, fine root length (average diameter < 0.5 mm), and number of root tips, were significantly (*p* < 0.001) increased in plants treated with BLACKJAK, compared to the untreated plants at all sampling times. At the molecular level, BLACKJAK treatment upregulated many of the evaluated genes. Moreover, both Real Time PCR and digital PCR showed that genes involved in hormonal response, such as PIN, ARF3, LOGL 10, GID1, and BRI1, were significantly (*p* < 0.05) upregulated by treatment with BLACKJAK. Our study provides essential information to understand the effect of a leonardite-based formulate on plant growth hormone metabolism, although the molecular and physiological basis for these complicated regulatory mechanisms deserve further investigations.

## 1. Introduction

Biostimulant compounds are a large and diversified class of compounds that contain one or more substances able to positively affect the crop quality when applied in small quantities to the soil or directly into the leaf surface. According to du Jardin [1], “a plant biostimulant is any substance or microrganism applied to plants with the aim to enhance nutrition efficiency, abiotic stress tolerance and/or crop quality traits, regardless of its nutrients content. By extension, plant biostimulants are also designated as commercial products containing mixtures of such substances and/or microrganisms”. These substances also stimulate rhizosphere microbes and soil enzyme activities [2,3]. Moreover, biostimulants show different beneficial effects on the production of hormones or growth regulators and photosynthetic processes [4,5]. Since their relatively recent introduction in agriculture, biostimulants do not belong to a proper class of chemicals, but they are still considered borderline substances between fertilizers and plant protection chemicals. 

Among biostimulant compounds, humic substances (HS) improve plant growth by metabolic changes, leading to an increase of crop yield [6]. The biostimulation mechanisms induced by HS are very complex. Some of them are hormone-like activities (auxin, gibberellin, or cytokine-like activity), stimulation of plant proton pumps (H^+^-ATPase) promoting root growth, increase of nutrient uptake, cell wall modification, and organ growth [7,8]. In wheat and maize, HS enhance lateral roots and improve seedling root growth [9,10], while in bean, rice, and pepper, HS improve tolerance to salt stress [11,12,13].

Leonardite is an oxidized form of lignite, very enriched in HS and characterized by well-known auxin-like effects [14,15]. Tahiri et al. [16] showed that HS extracted from leonardite influenced the whole-plant physiology by a transcriptional mechanism of regulation associated with a hormone-like activity, as well as changes in nitrogen and carbon metabolisms in *Betula pendula* and *Alnus glutinosa*. Leonardite application has been shown to improve Fe nutrition and N and P uptake, enhancing root growth and crop yield [17]. Moreover, Akinremi et al. [18] found that leonardite increased canola yield by supplying S directly and facilitating the uptake of other nutrients.

Sugar beet (*Beta vulgaris* L.) is a root crop mainly growing in temperate climates and providing about 20% of the world’s annual sugar production [19]. Plants are highly susceptible to water and nutritional stresses during the first stage of growth [20]. The application of biostimulants in this critical stage could accelerate root growth, allowing a rapid colonization of the soil and, therefore, water and nutrient uptake. To this purpose, Barone et al. [21] found that extracts from microalgae, *Chlorella vulgaris* and *Scenedesmus quadricauda*, are promising biostimulants in the early stages of sugar beet cultivation. Extracts from these two algae increased nutrient acquisition and root growth. Moreover, at a molecular level, the use of a select gene panel allowed the study of differences in genetic expression of plants treated or untreated with the two microalgae extracts. The same gene panel has been sucessfully used to evaluate sugar beet responses to changes in sulfate availability [22].

Considering the worldwide use of biostimulants, their increasing application in agriculture, and their different action mechanisms in a variety of plant species, in-depth molecular investigation of plant response to their application is one of the main points involved in this study. Particularly, complex effects of HS activity on sugar beet are poorly understood and need rigorous evaluation. In this work, we evaluated the responses of root traits and the expression of the nutrition-related gene panel to the application of a leonardite-based biostimulant in sugar beet grown in Hoagland’s solution under controlled conditions. 

## 2. Results

### 2.1. Chemical Characterization of Biostimulant

Table 1 shows the elemental composition of BLACKJAK, analyzed by combustion and ICP analysis. 

As expected, the most abundant element is C, which amounts to 51.1% of all matter, followed by Fe (2.11%), Na (1.27%), Al (1.02%), Ca (0.90%), N (0.88%), S (0.75%), K and Mg (0.15%), and P (0.06%). Table 2 shows the composition of C functional groups (%), estimated from relative integration values of ^13^C CPMAS NMR spectra of BLACKJAK. 

It is characterized by a balanced content of alkyl and aromatic carbon, while the N and O alkyl carbon content was lower (Table 2). As highlighted in the NMR spectrum (Figure 1), the alkyl area is dominated by three peaks at 19, 33, and 56 ppm, respectively.

The first peak is due to the terminal methyl carbon from either short-chain hydrocarbons or long-chain n-alkanes [23], the second one indicates the presence of long-chain polymethylene structures [24], and the last one is referred to as a methoxylic carbon structure [25]. The aromatic area of spectrum is characterized by three principal peaks, one peak at 128 ppm indicating the presence of a substitute and alkyl-unsubstituted aromatic carbon [26] and the other two near peaks at 143 and 146 ppm, respectively, which are attributed to the phenolic carbons [27]. The peaks observed in the other two regions of the spectrum analyzed are less evident than the principal peaks. The O-alkyl region contains a resonance band at 68 ppm, attributed to the presence of sugars, cellulose, alcohols, and/or ethers [24], and the carboxyl-like carbon region shows a major peak at 172 ppm, indicating the presence of either a carboxylic, an ester, or an amide carbon [27]. Finally, the degree of hydrophobicity calculated for BLACKJAK was found to be 3.1.

### 2.2. Ionomic Analysis

The contents of B, Ca, Cu, Fe, K, Mg, Mn, P, S, and Zn of the Hoagland solution and seedling roots were determined by ionomic analysis at the beginning and at the end of the test. Elemental composition of the Hoagland solution was not altered by BLACKJAK treatment due to the high dilution rate (1:1000; 0.5 mg C L^−1^). The roots’ elemental composition was significantly changed by treatment, as shown in Table 3. 

Principal component analysis of mineral elements revealed that untreated and treated root samples belonged to separate clusters (Figure 2). 

Factors 1 and 2 explained 56.22% and 40.82% of the total variation, respectively. The major elements that contributed to factor 1 were Mg and P, while Zn and Cu resulted in dominating factor 2.

### 2.3. Root Morphological Analysis 

Root length and root surface area significantly (*p* < 0.05) increased in plants treated with BLACKJAK, compared to the untreated plants at all sampling times (24 h, 48 h, and 72 h) (Figure 3). 

Immediately after 24 h of treatment, the length and the surface area greatly increased in comparison to the untreated samples by 33% and 24%, respectively. The greatest effect of the treatment was reached at 72 h, being the length and the surface area of roots at 37% and 34% greater than the control, respectively. Plants treated with BLACKJAK showed a higher number of tips, forks, and crossings as compared to the untreated plants (Figure 3). After 72 h of treatment, the number of tips, forks, and fine roots dramatically increased by 40%, 56%, and 40%, respectively. Moreover, in treated plants at 72 h, the production of fine roots ranging from 0 to 0.5 cm was strongly enhanced (40%) (Figure 3). 

### 2.4. Molecular Analysis

In order to understand the molecular response in sugar beet root treated with BLACKJAK, the transcript levels of 53 genes involved in important plant processes, including metabolic pathways and hormonal metabolism, were analyzed by Open Array Real Time PCR and further validated by digital PCR. 

The expression level of 53 genes, putatively involved in nutrient metabolism, was evaluated in untreated and treated plants. The ANOVA showed a significant effect of treatment (*p* < 0.01), time of exposition (*p* < 0.01), and gene (*p* < 0.05), as reported in Table 4.

Most of the evaluated genes were significantly upregulated by the treatments. The exposure time effect was significant and 48 h showed, on average, higher relative values than 72 h. Particularly, BLACKJAK treatment upregulated the expression of genes involved in nutrient acquisition, such as sulfotransferase (AIX02JA) and inorganic phosphate transporter (AILJKS3 and AIMSIZB) (Figure 4). Two genes, AI0IYVQ and AIAA083, showed amplification problems for one and two of the exposure times, respectively. The problem could be related to primer annealing specificity.

Among the 53 genes, we focused our attention on five genes related to hormonal response: PIN (auxin efflux component 1, AI1RW1X), ARF3 (auxin response factor 3, AIBJZFA), LOGL 10 (cytokinin riboside 5-monophosphate phosphoribo hydrolase, AII1OGA), GID1 (gibberelin receptor, AI20L89), and BRI1 (brassinosteroid insensitive 1, AI0IYVP). Particularly, Real Time PCR results showed that PIN expression at 24 h keeps a value corresponding to the basal levels (untreated sample), whereas it rapidly decreases in comparison to the untreated in correspondence to 48 h of treatment (Figure 4).

Finally, PIN transcripts sharply increased after 72 h of treatment, reaching a value 0.58 times higher than the untreated sample. A similar behaviour was observed by ARF3, where the transcript level increased after 72 h of treatment, with a value 0.47 times higher than the untreated sample. LOGL 10 after 24 h immediately increased, reaching a value 0.57 times higher than the untreated sample, whereas at 48 h, LOGL 10 expression reached values corresponding to the basal levels of the untreated sample. Finally, it rapidly decreased below that of the untreated sample after 72 h of treatment with BLACKJAK. The effect of BLACKJAK on the expression of GID1 was weaker than that observed on PIN, ARF3, and LOGL 10 expression, although the GID1 overall expression trend was rather similar to the PIN and ARF3 expression patterns (Figure 4). In fact, similarly to PIN and ARF3 expression, GID1 expression recorded a weak increase after 24 h of treatment, followed by a decrease of expression level in comparison to the untreated samples after 48 h of treatment. Finally, GID1 expression levels increased after 72 h of BLACKJAK treatment. The effect of BLACKJAK on the expression of BRI1 after 24 h corresponded to the basal level and rapidly increased after 48 h and 72 h, reaching a relative expression value of 0.55 higher than the untreated sample. To further assess accuracy of relative expression values obtained by Real Time PCR, the transcript levels of PIN, ARF3, LOGL 10, GID1, and BRI1 were also analyzed by digital PCR. Gene quantification values, reported as copies μL^−1^, obtained by digital PCR are shown in Table 5.

Results of quantitation obtained from Real Time PCR and dPCR results were significantly correlated as determined by the correlation coefficients of r^2^ = 0.977 for PIN, r^2^ = 0.981 for ARF3, r^2^ = 0.991 for LOGL 10, r^2^ = 0.995 for GID1, and r^2^ = 0.975 for BRI1. Scatter plots obtained by QuantStudio 3D Analysis Suite Cloud Software using the relative quantitation application are represented in Figure 5, Figure 6, Figure 7, Figure 8 and Figure 9. 

Wells with yellow dots indicated ROX signal (passive reference), while wells with red dots indicated VIC signal, correlating with the presence and quantity of targeted genes. QuantStudio 3D Digital PCR Analysis Suite Cloud Software calculated the absolute number of targets and performed statistics, providing the confidence interval and precision. Precision is defined as the spread of confidence interval around two sample concentrations at a given confidence interval. The overall precision of the chips analysed was 5%. Red dots are described as the number of dots converted by the software into copies μL^−1^ of target. 

A scatter plot of the PIN gene is presented in Figure 5, showing the presence of 102.6 (Figure 5a), 80.90 (Figure 5b), and 129.07 (Figure 5c) copies μL^−1^ of the target, in plants after 24, 48, and 72 h of treatment, respectively. Similarly, the scatter plot of ARF3 (Figure 6) shows the presence of 105.09 (Figure 6a), 90.58 (Figure 6b), and 132.19 (Figure 6c) copies μL^−1^ of the target. Scatter plots for LOGL 10 and GID1 are reported in Figure 7 and Figure 8, respectively, showing 131.95, 119.34, and 108.98 copies μL^−1^ of LOGL 10 and 121.02, 110.12, and 123.02 copies μL^−1^ of GID1, both in plants after 24, 48, and 72 h of treatments. BRI1 gene is characterized by the presence of 109.98, 128.87, and 133.98 copies μL^−1^ in plants treated for 24, 48, and 72 h, respectively (Figure 9a–c).

## 3. Discussion

The effect of HS on plant growth is multi-faceted, resulting in a final increase of yield. This increase starts from morphological variations of the root system, governed by changes in the metabolism, which in turn are induced by gene expression modifications linked to the assimilation of nutrients as well as the synthesis and the transport of hormones. 

BLACKJAK content of C, N, S, and P (Table 1) is similar for other humic substances obtained from leonardite [28,29]. Its characterization seems to highlight the presence of a relationship between its composition and the morphologic response in sugar beet roots. This relationship between structure of the formulate and its bioactivity was successfully shown by Piccolo et al. [30], using ^13^C CPMAS NMR spectra in humic acid isolated from different sources. They also found that the degree of hydrophobicity of humic substances is closely related to the increase of root growth. Our results show a degree of hydrophobicity value greater than those observed for HAs of a different origin, ranging between 0.61 and 2.87 [31], and intermediate between those calculated for humic substances extracted from leonardite. These values are 1.08 for humic acids from compost [32] and 4.75 for humic substances from leonardite, as referenced by the International Humic Substances Society [33]. Specifically, methoxilic groups, aryl groups, and carboxylic acids seem to be responsible for humic acid bioactivity. Moreover, a complex humic structure showing lower functionalization (unsubstituted aromatic and aliphatic groups) was closely related to the growth of larger roots, whereas less complex humic substances with functionalized structures (O, N substituted aliphatic chains) were related to the root length and the initiation of new smaller roots [33]. BLACKJAK characterization by ^13^C CPMAS NMR and ionomic analyses (Figure 1, Table 1 and Table 2) was perfectly compatible with the effect on both large and small roots, as also confirmed by morphological traits in sugar beet roots (Figure 3). According to Canellas and Olivares [34], the exogenous application of HS induced a greater growth response in monocotyledons than in dicotyledons. Nevertheless, our results suggest that BLACKJAK may also be a very promising biostimulant in dicotyledons.

Humic substances affect plants primary metabolism by effecting gene expression. Particularly, humic substance activity influences glycolysis and Krebs cycle, nitrate metabolism, and photosynthesis [35]. This has been widely demonstrated on maize by Nardi et al. [36]. 

In this work, we evaluated the responses of genes and root traits related to soil exploration and nutrient uptake on the application of a leonardite-based biostimulant (BLACKJAK) in sugar beet grown in Hoagland’s solution under controlled conditions. Plant root apparatus is shaped by the interaction between genes and environment. Roots maintain their morphology according to their genotype, but many differences can be attributed to the environment in which they grow [37]. Biostimulants are able to modify root growth and architecture [38]. Our results successfully show that the morphology of root apparatus was positively affected by the treatment with BLACKJAK when added to the Hoagland solution (Figure 3). These results are particularly interesting in sugar beet, since rooting is related to the increase of water-nutrient uptake and plant growth, hence the improving of final sugar yield [39]. Root morphological traits after 24 h of treatment seems to be regulated by cytokinins, which affect the regulation of root extension [40]. Root parameters, such as length and surface (Figure 3), increase only after 24 h, corresponding to the highest value of LOGL 10 expression (Table 5). After 72 h, the expression levels of PIN, ARF3, GID1, and BRI1 increased and LOGL 10 expression level decreased (Table 5). The number of tips, forks, and fine roots dramatically increased (Figure 3), suggesting that auxin can stimulate cell division in the root pericycle, crucial for the initiation and elongation of adventitious roots [41]. Our data are also in accordance with the results obtained in rice, showing that PIN proteins play an important role in adventitious root emergence and root branching [42]. In fact, PIN-mediated auxin transport is necessary in the early embryogenesis stage of plant body development [43]. Similarly in *Arabidopsis*, although different PINs may be characterized by different auxin efflux activities, root-hair-specific expression of PINs was strictly correlated to the increase of root growth, probably being related to an enhanced cell elongation [44].

In parallel with phenomics, a molecular genomics approach allows for the deep study of the effect of the biostimulant products. Here, gene expression changes have been quantified through Real Time PCR after BLACKJAK treatment. Most of the 53 analysed genes changed their expression pattern (Figure 4). The evaluated set of 53 genes was chosen since they were induced by sulfate availability and microalgae treatments in previous experiments [21,22]. The genes analysed here are related to many metabolic functions and putatively influenced by imposed treatment. Particularly, both Real Time PCR and digital PCR showed that genes involved in hormonal response, such as PIN, ARF3, LOGL 10, GID1, and BRI1, were significantly upregulated by treatment with BLACKJAK. 

Real Time PCR and digital PCR together are valuable techniques for evaluation of gene expression responses to biostimulant application [45]. For example, Povero et al. [46] selected a set of tomato genes involved in different biochemical pathway in order to screen the effect of several seaweed-based prototypes using qPCR. Particularly, dPCR provided higher sensitivity and reproducibility than Real Time PCR for a low abundant target and to reduce the effects of PCR inhibitors [47]. In this context, Cremonesi et al. [48] obtained higher sensitivity using dPCR compared to Real Time PCR to quantify common foodborne pathogens in food matrices. In the present study, using high quality cDNA, inhibition was not observed in Real Time PCR and gene expression results showed correlation with dPCR values. Similar results have been found by Alikian et al. [49], comparing both techniques for the evaluation of residual disease in chronic myeloid leukemia. Another comparison of the two PCR methods was done by Kinz et al. [50], obtaining a linear regression value of 0.9983 between Real Time PCR and dPCR.

To obtain a complete picture of gene expression changes within given experimental conditions, microarray and/or high-throughput Real Time PCR are highly suggested [51]. Among these approaches, the OpenArray technology adopted here resulted as expected, not only as a validation tool but also considering its feasibility to the experimental design. In fact, target genes and number of samples can be adjusted according to the experimental design [52]. Here, we effectively adopted OpenArray technology to achieve rigorous evaluations of complex effects of BLACKJAK on sugar beet. As demonstrated previously by Barone et al. [21], nutrition-related genes were induced since BLACKJAK increases root development (Figure 3) and, as a consequence, nutrient uptake. In fact, BLACKJAK treatment induced the upregulation of the genes linked to sulfur metabolism and phosphate transport as well as genes involved in cell organization (Figure 4). In this study, we focused our attention on five genes involved in hormonal responses in order to investigate in depth the hormone-like effect of BLACKJAK (Table 5, Figure 5, Figure 6, Figure 7, Figure 8 and Figure 9). Hormones also regulate plant physiological processes in the initial stage of plant growth [53]. During the initiation of the lateral root primordial, auxin stimulate and cytokinin inhibit the cell division in the root pericycle, respectively, increasing and reducing their numbers [54]. In rice, lateral root development depends on cytokinin action, which has an inhibitory effect on lateral root initiation, but a great stimulatory effect on their emergence and elongation [40]. Moreover, Rani Debi et al. [40] demonstrated that the addition of different concentrations of auxin to the growth solution counteracted the inhibitory effect of cytokinin on lateral root formation. However, at increasing auxin concentrations, the stimulatory effect on lateral root elongation of cytokinin was gradually reduced. In addition, the expression of auxin response genes is regulated by members of the auxin response factor (ARF) family. ARF genes regulate different plant developmental stages through proteins associated with DNA binding, transcriptional activation, or repression [55]. Phosphorylation of some ARF genes by brassinosteroid-insensitive genes has been demonstrated to enhance auxin signaling during lateral root formation [55]. Finally, the GA-GID1-DELLA system, the major repressors of gibberellin expression, plays a crucial role in the regulation of root extension growth with auxin [56]. Root cell differentiation is also controlled by brassinosteroids, specifically during primary root formation when brassinosteroid insensitive 1 is highly expressed [57].

Our results may suggest that during the first 24 h of treatment, LOGL 10 may be involved in the biochemical event leading to the early morphological changes, whereas after 72 h, LOGL 10 expression decreased and PIN, ARF3, GID1, and BRI1 expression increased (Table 3 and Table 5). These latter genes are probably responsible for the greatest effect on root morphological traits reached at 72 h from treatment.

## 4. Materials and Methods

### 4.1. Chemical Characterization of HS

The new leonardite-formulate used in this work was a liquid commercial product patent-pending, hereafter named BLACKJAK and provided by Sipcam SpA (Italy).

#### 4.1.1. Elemental Analysis

BLACKJAK was analyzed by combustion (Elementar vario MACRO CNS, Elementar Analysensesystemse GmbH, Germany) for C, N, and S content and by ionomic analysis. Samples were digested using a concentrated HNO_3_ solution in a microwave system. The element concentration of K, P, Ca, Mg, Na, Fe, and Al was determined by inductively coupled plasma ICP-OES, Ciros Vision EOP (Spectro A. I. GmbH, Germany). Elements were quantified using certified multi-element standards, following the procedure previously adopted by Stevanato et al. [58]. 

#### 4.1.2. NMR Analysis

NMR analysis of BLACKJAK was performed by solid-state ^13^C MAS NMR spectra, fully proton-decoupled using a Bruker Avance II 400 MHz instrument (Bruker Corp., USA) and operating at 100.63 MHz. Samples of about 50 mg of the freeze-dried material were used to fill the rotors (7 mm diameter) using a spinning rate of 8000 Hz s^−1^. The parameters used in the analysis were as follows: Spectral width 20,000 Hz, data points 2 K, 100,000 scans, 5 μs, 90° of excitation pulse, and 4 s of relaxation delay. The HPDEC pulse sequence was of 300 W (9H) decoupling power. The FID was zero-filled and processed with 5 Hz line broadening. Spectra and the distribution of the diverse forms of carbon were designed according to the area of the different NMR spectrum regions, as indicated by Baglieri et al. [59] (aliphatic, 0 to 45 ppm; N and O alkyl, 45 to 95 ppm; aromatic, 95 to 160 ppm; carboxyl, 160 to 195 ppm). The degree of hydrophobicity was calculated according to Baglieri et al. [59] as: HB/HI = [(0 − 45) + (95 − 160)/(45 − 95) + (160 − 195)]. 

### 4.2. Experimental Design, Plant Material, and Growing Conditions

#### 4.2.1. Preliminary Investigation

In order to select the appropriate BLACKJAK dose, two preliminary tests were conducted. The first test was performed by using the following BLACKJAK dilutions: 1:10, 1:100, 1:1000, and 1:10000. The parameter measured was the root length (after 72 h of treatment) in seedlings grown as described in the next paragraph. The highest value of root length was recorded by using BLACKJAK 1:1000 (data not shown). To validate this result, the second test was performed in the same experimental condition in a short range (around 1:1000) by using the following dilutions: 1:700, 1:1000, 1:1300, and 1:1600. In the second experiment, the following root parameters (length, surface area, tips, and forks) were measured by WinRHIZO software as described in the paragraph below (data not shown). In this validation trial, the best dilution was BLACKJAK 1:1000 as well. In the controls, the BLACKJAK was replaced with water.

#### 4.2.2. Experimental Design

Seedlings of sugar beet were grown under controlled conditions in Hoagland’s solution containing 200 μM Ca(NO_3_)_2_, 200 μM KNO_3_, 200 μM MgCl_2_, 40 μM KH_2_PO_4_, 10 μM FeNaEDTA, 4.6 μM H_3_BO_3_, 910 nM MnCl_2_, 86 nM ZnCl_2_, 36 nM CuCl_2_, and 11 nM NaMoO_4_. After five days of growth, an aliquot of the concentrated BLACKJAK was added to the solution to obtain a final dilution of 1:1000 (0.5 mg C L^−1^). Sixty seedlings per replicate were collected immediately and 24, 48, and 72 h after the treatment. All treatments were done in triplicate and the experiment was performed five times.

#### 4.2.3. Plant Material 

This study included the sugar beet hybrid “Variety_1” from the Department of Agronomy, Food, Natural resources, Animals and Environment-DAFNAE (University of Padova, Italy) collection. This hybrid was previously genotyped as part of a fingerprinting study [60]. It is diploid, multigerm, and homozygous for resistance to rhizomania (Rz1). Seeds were produced in 2017 and calibrated at a diameter of 4 mm.

#### 4.2.4. Growing Conditions

Seeds were soaked in 76% ethanol for 5 min, rinsed with sterilized water, and distributed on distilled water-moistened filter paper to germinate. Germination was carried out in a growth chamber at 25 °C in the dark. Germinated seedlings were transplanted in a 500 mL pot containing sterile Hoagland solution, according to Arnon and Hoagland [61], with a density of 30 plants per pot. Plants were cultivated in a climatic chamber at 25/18 °C and 70/90% relative humidity with a 14/10 h light/dark cycle (PPFD above shoot: 300 μE m^−2^ s^−1^). Five days after germination, 0.5 μL of BLACKJAK (dilution 1:1000; 0.5 mg C L^−1^) was added to the nutrient solution (0.5 μL of water in the blanks). 

### 4.3. Ionomic Analysis

The element concentration of the Hoagland solution and seedlings, at the beginning and at the end of the test, was determined by ionomic analysis. Root samples were digested with concentrated HNO_3_ in a microwave system. The element concentration was determined by inductively coupled plasma ICP-OES. Elements were quantified using certified multi-element standards. 

### 4.4. Root Morphological Analysis 

Root morphological parameters, such as total root length, surface area, total number of tips, forks, crossings, and fine and medium root length, were determined 24, 48, and 72 h from the treatment by computerized scanning (STD 1600, Regent Instruments, Canada) and analyzed using WinRHIZO software (Regent Instruments). 

### 4.5. Molecular Analysis

#### 4.5.1. Root RNA Isolation and cDNA Synthesis

Total RNA was extracted from 0.2 g of root tissues using a eurogold trifast kit (Euroclone, Milan, Italy) and DNase-digested with turbo DNA-free kit (Thermo Fisher Scientific, MA, USA), according to manufacturer recommendations. The RNA was quantified by absorbance at 260 nm and RNA quality was determined using agilent 2100 bioanalyzer (Agilent Technologies, USA). Reverse transcription of RNA (500 ng) was achieved using the superscript III reverse transcriptase (Thermo Fisher Scientific), following the manufacturer’s instructions. 

#### 4.5.2. Real Time PCR and Digital PCR Analysis 

The transcript level of 53 genes related to important morphophysiological processes in sugar beet was analysed by Real Time PCR on the QuantStudio 12K Flex Real Time PCR System (Life Technologies, USA), using 2.5 μL of 2× TaqMan Open Array master mix (Life Technologies, USA) and 2.5 μL of cDNA. The thermocycler program consisted of 10 min of pre-incubation at 95 °C, followed by 50 cycles of 15 s at 95 °C and 1 min at 60 °C. Primer and probe sequences are reported as Appendix A. The comparative threshold Ct method was used to analyze the genes’ relative expression. Data were normalized against the average transcript abundance of three housekeeping genes (*Tubulin*, Bv2_037220_rayf; *GAPDH*, Bv5_107870_ygnn; *Histone H3*, Bv6_127000_pera). The fold change in the expression of genes was calculated using Formula 2^−ΔΔCt^, where ΔΔCt = (Ct target gene − average Ct reference genes) treatment − (Ct target gene − average Ct reference genes) control. Data are the mean of three biological replicates, each one composed of three technical replicates ± SE of one representative experiment. The Ct method was used to quantify the relative gene expression levels and the results were expressed as 2^−ΔCt^, where ΔCt = (Ct of reference gene − Ct of target gene).

Digital PCR was conducted using the QuantStudio 3D (Thermo Fisher Scientific) with the following conditions: 96 °C for 10 min, 39 cycles at 60 °C for 2 min, and at 98 °C for 30 s, followed by a final extension step at 60 °C for 2 min. The PCR mix was composed of 8 μL of QuantStudio 3D Digital PCR Master Mix (Thermo Fisher Scientific, USA), 1.44 μL of both forward and reverse primers, 0.4 μL of FAM probe, and 2.82 μL of nuclease-free water. Digital PCR data were analysed with the QuantStudio 3D AnalysisSuite Cloud software (Thermo Fisher Scientific, USA). The absolute levels of the target gene were expressed as the number of copies per microgram of RNA. Confidence interval and precision of dPCR analysis were calculated using Poisson statistics directly by the QuantStudio 3D AnalysisSuite Cloud software.

### 4.6. Statistical Analysis 

A normality test (Kolmogorov–Smirnov) and homogeneity of variance test (Levene’s test) were performed. One-way ANOVA analysis was carried out with the least significant difference test (LSD) to determine whether untreated and treated samples differed in terms of evaluated variables. In case of significant difference (*p* value < 0.05), means were separated using the LSD test. PCA analysis was conducted for data analysis with the aim to identify chemical differences between untreated and treated samples. Variables were presented as mean and standard error of the mean (standard error). All statistical analyses were done using Statistica software v. 13.4 (TIBCO Software, USA).

## 5. Conclusions

These preliminary results appear to be very interesting and promising, since the application of BLACKJAK to a sensitive crop such as sugar beet sets the stage for a great impact of this new formulate in different kinds of crop production. It is worth noting that m less information is available on plant physiological responses to HS isolated from brown coal such as leonardite, compared to HS isolated from other sources. In fact, the genes analyzed in the present study represent many metabolic functions strictly related to the development of the root apparatus of sugar beet, which change their expression in the early stage of exposure to the BLACKJAK treatment. Nevertheless, we have chosen to show only a frame of the plants’ response, aiming to evaluate the BLACKJAK biostimulant effect. Moreover, our study might also provide essential information to understand the effect of HS from leonardite on plant growth hormonal metabolism, although the molecular and physiological basis for these complicated regulatory mechanisms deserve further investigation. Moreover, the method proposed here could be used and adapted to study the phenotypic and molecular effect of different products, since the biostimulants sector is growing exponentially.

## Figures and Tables

**Figure 1 plants-08-00181-f001:**
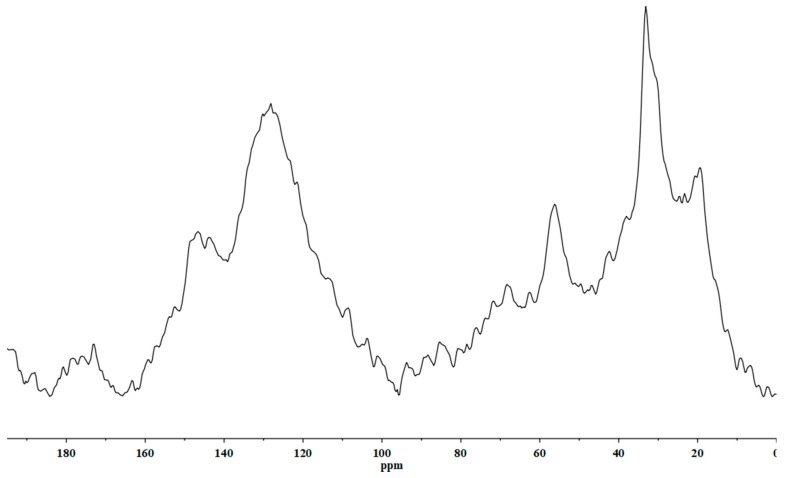
Spectrum of ^13^C CPMAS NMR of BLACKJAK.

**Figure 2 plants-08-00181-f002:**
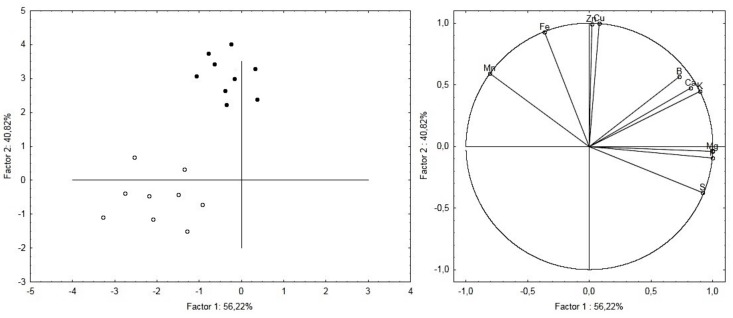
Principal components analysis (PCA) of the roots ionome. The figure on the Left shows the modification of the roots ionome as a function of the treatment. Filled dots show treated samples whereas empty dots show untreated root samples.The figure on the Right shows the relationship between variables and principal components and also highlights relationships between the variables themselves.

**Figure 3 plants-08-00181-f003:**
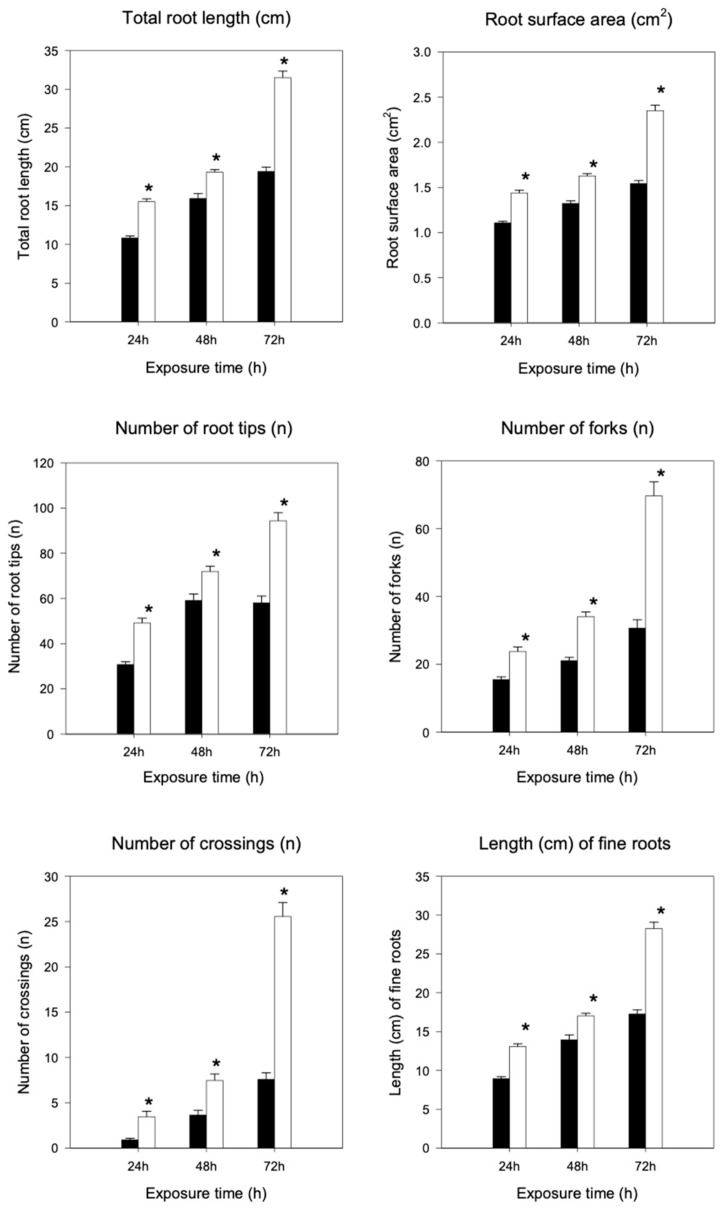
Total root length, root surface area, number of root tips, forks, crossings, and length of fine roots of untreated plants (black) and BLACKJAK treated plants (white) after 24 h, 48 h, and 72 h. Error bars indicate standard error. The values are means of data from 75 seedlings. * shows significant difference at *p* < 0.05 level, compared to the control.

**Figure 4 plants-08-00181-f004:**
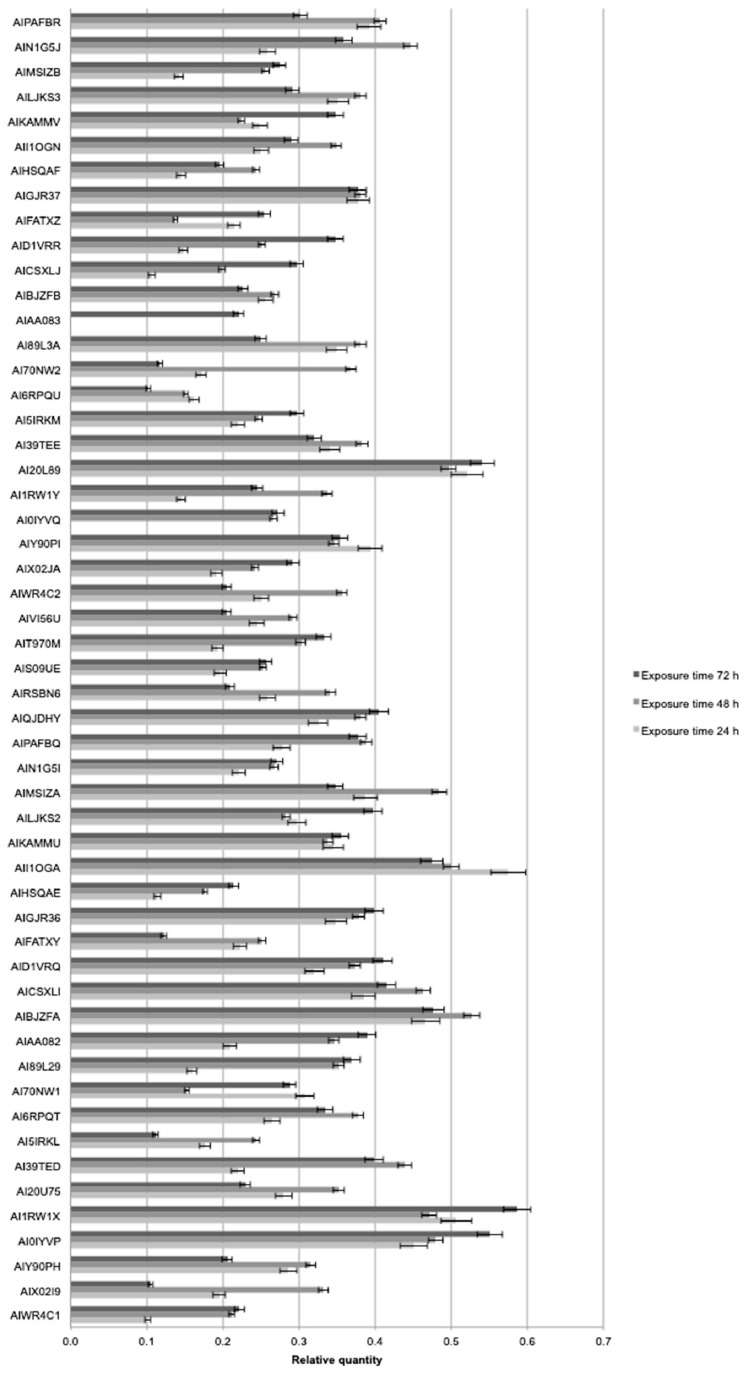
Relative expression value of the 53 genes evaluated in sugar beet root after 24 h, 48 h, and 72 h of treatment with BLACKJAK.

**Figure 5 plants-08-00181-f005:**
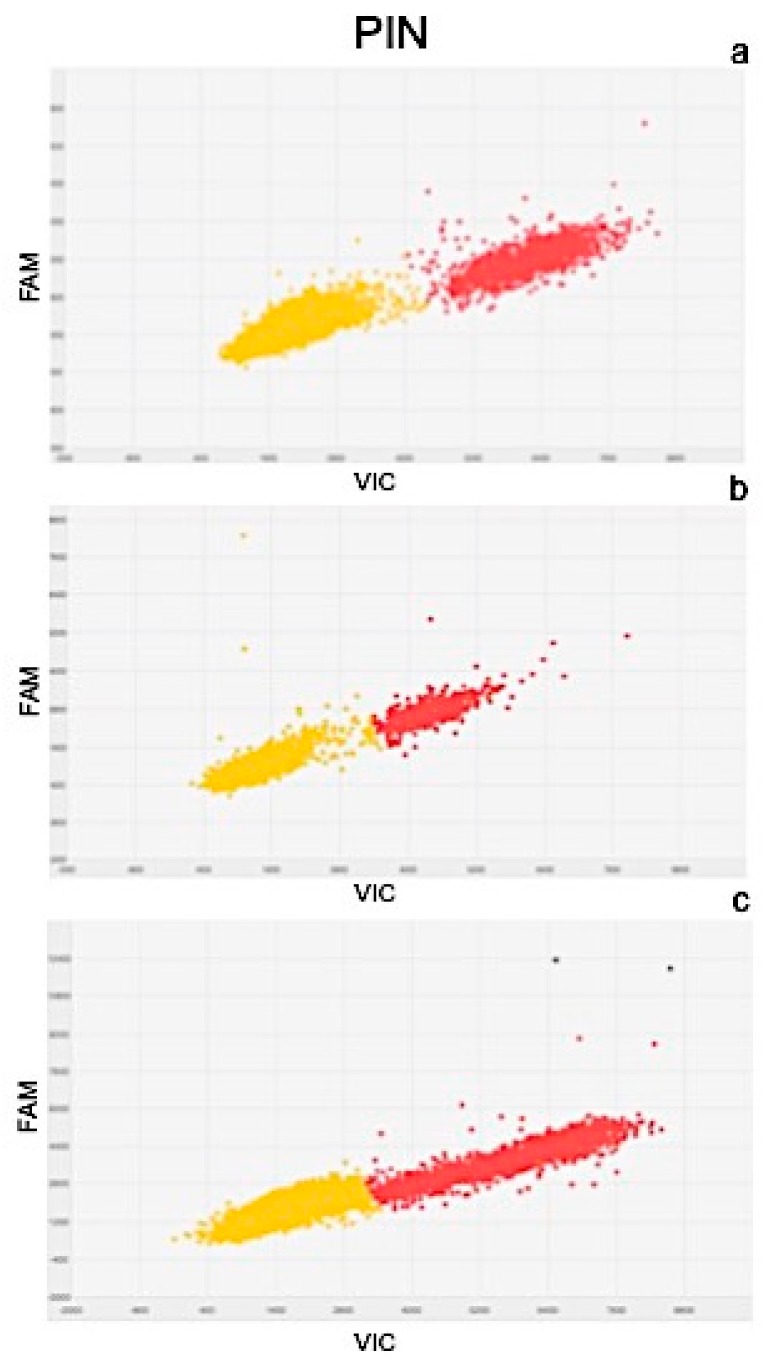
Scatter plots of PIN gene, obtained by QuantStudio 3D Analysis Suite Cloud Software using the relative quantitation application. Scatter plot analysis was done on samples after (**a**) 24, (**b**) 48, and (**c**) 72 h after treatment.

**Figure 6 plants-08-00181-f006:**
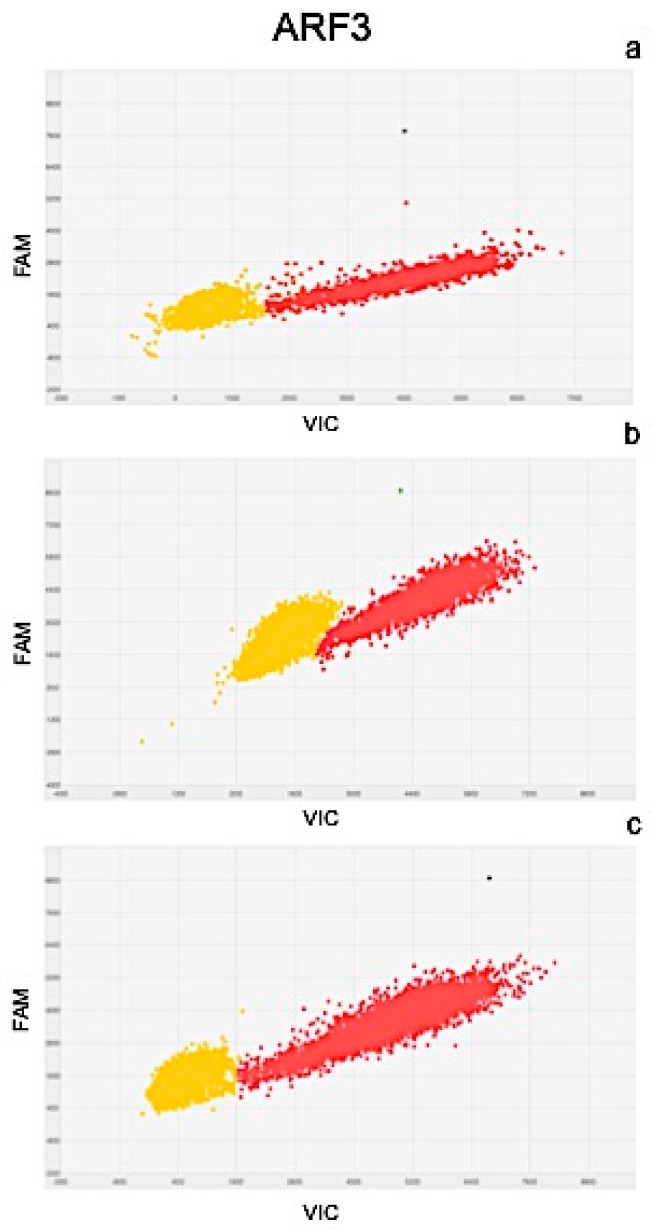
Scatter plots of ARF3 gene, obtained by QuantStudio 3D Analysis Suite Cloud Software using the relative quantitation application. Scatter plot analysis was done on samples after (**a**) 24, (**b**) 48, and (**c**) 72 h after treatment.

**Figure 7 plants-08-00181-f007:**
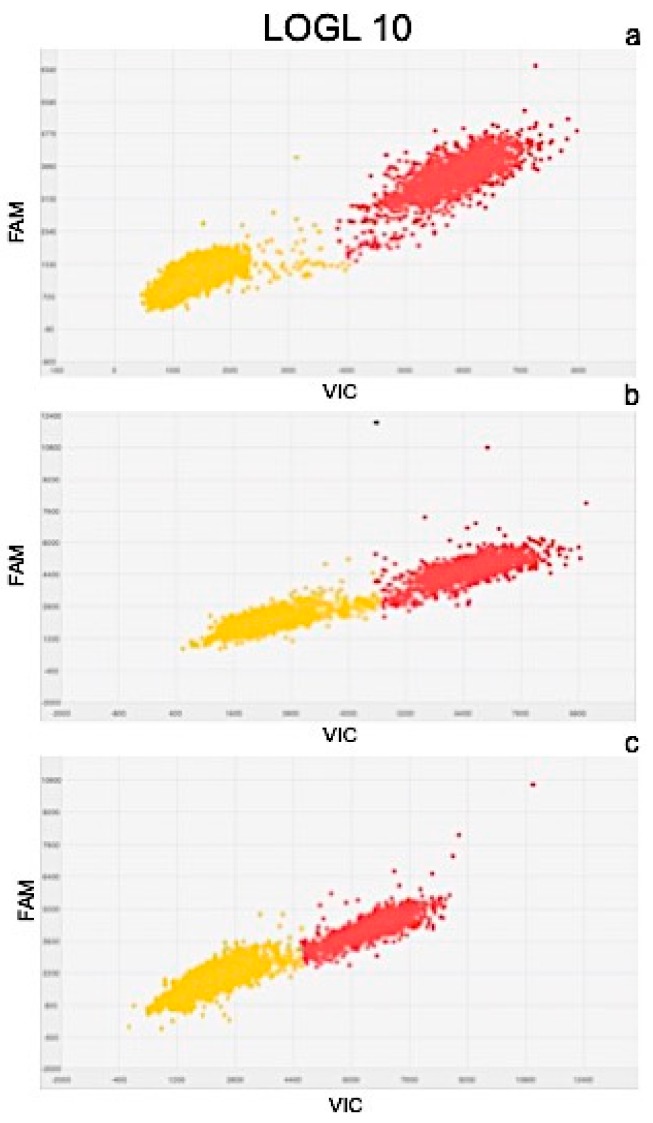
Scatter plots of LOGL 10 gene, obtained by QuantStudio 3D Analysis Suite Cloud Software using the relative quantitation application. Scatter plot analysis was done on samples after (**a**) 24, (**b**) 48, and (**c**) 72 h after treatment.

**Figure 8 plants-08-00181-f008:**
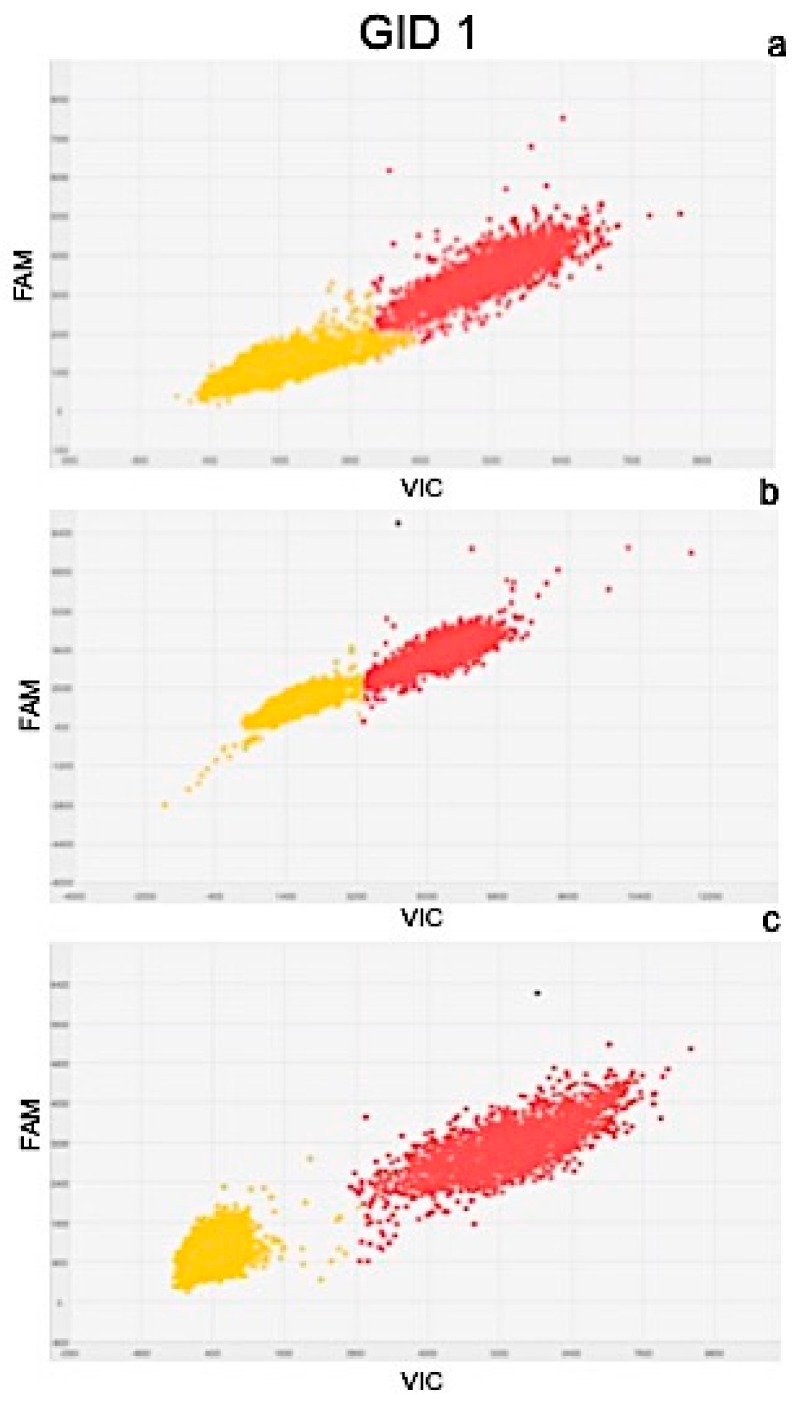
Scatter plots of GID1 gene, obtained by QuantStudio 3D Analysis Suite Cloud Software using the relative quantitation application. Scatter plot analysis was done on samples after (**a**) 24, (**b**) 48, and (**c**) 72 h after treatment.

**Figure 9 plants-08-00181-f009:**
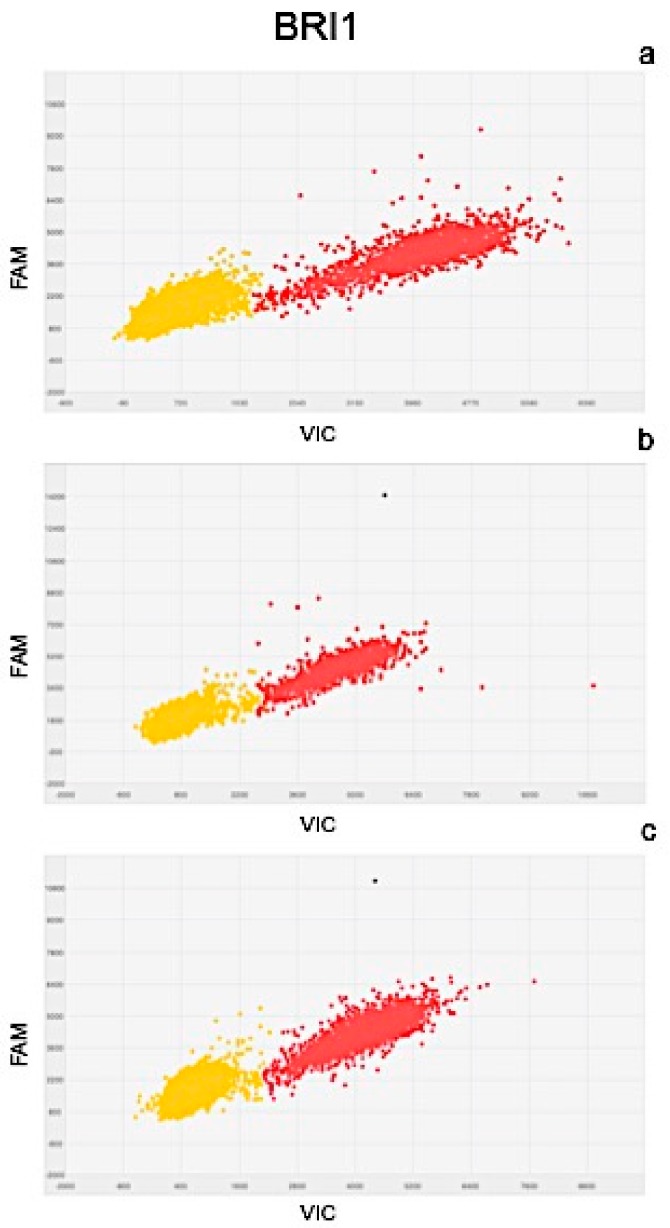
Scatter plots of BRI1 gene, obtained by QuantStudio 3D Analysis Suite Cloud Software using the relative quantitation application. Scatter plot analysis was done on samples after (**a**) 24, (**b**) 48, and (**c**) 72 h after treatment.

**Table 1 plants-08-00181-t001:** Elemental composition of BLACKJAK determined by combustion and ICP analysis.

Element	Method	Percentage (%)
C	Combustion analysis	51.1
N		0.88
S		0.75
K	ICP analysis	0.15
P		0.06
Ca		0.90
Mg		0.15
Na		1.27
Fe		2.11
Al		1.02

**Table 2 plants-08-00181-t002:** Composition of C functional groups (%) estimated from ^13^C CPMAS NMR spectra of BLACKJAK.

C Functional Group	Method	Percentage (%)
Alkyl-C	NMR analysis	36.1
N and O-alkyl-C		20.9
Aromatic-C		39.7
Carboxyl-C		3.3

**Table 3 plants-08-00181-t003:** Elemental composition of untreated and treated seedling roots. Data are expressed as mg g^−1^ DM.

Element	Untreated Roots			Treated Roots			*t*-Test
B	2.9	±	0.5	5.2	±	0.2	*p* < 0.001
Ca	151.8	±	10.8	234.5	±	13.9	*p* < 0.01
Cu	1.1	±	0.02	2.9	±	0.1	*p* < 0.001
Fe	52	±	6.2	112	±	4.6	*p* < 0.001
K	2014.4	±	192.4	3959.3	±	234.3	*p* < 0.001
Mg	182.2	±	7.8	545.9	±	11.6	*p* < 0.001
Mn	6.7	±	0.5	7.2	±	1.2	*p* < 0.05
P	554.2	±	34.1	937.8	±	43	*p* < 0.01
S	277.8	±	12.4	332.8	±	26.1	*p* < 0.05
Zn	20.4	±	3.4	51.2	±	9.8	*p* < 0.001

**Table 4 plants-08-00181-t004:** Analysis of variance (ANOVA) showing the effect of different treatments, time of exposition, and genes (* *p* < 0.05; ** *p* < 0.01, factorial ANOVA test) on the expression of 53 sugar beet genes putatively involved in nutrient metabolism.

Effect	df	SS	MS	F	*p*
Treatment	1	1421	1421	115.2	**
Exposure	2	2723	1362	231.3	**
Gene	52	4523	87	6.9	*

**Table 5 plants-08-00181-t005:** Gene expression values, reported as copies μL^−1^, of genes encoding for PIN, ARF3, LOGL 10, GID1, and BRI1 in sugar beet root after 24 h, 48 h, and 72 h of treatment with BLACKJAK.

Gene Expression Values (copies μL^−1^)	Exposure Time					
	24 h		48 h		72 h	
PIN	102.06	± 3.2	80.9	± 4.7	129.07	± 6.3
ARF3	105.09	± 2.5	90.58	± 3.1	132.19	± 3.5
LOGL 10	131.95	± 1.9	119.34	± 2.4	108.98	± 2.0
GID1	121.02	± 4.1	110.12	± 1.5	123.02	± 4.2
BRI1	109.98	± 1.2	128.87	± 2.3	133.02	± 3.4

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
