# Peer review of "Molecular and Morphological Changes Induced by Leonardite-based Biostimulant in Beta vulgaris L."

_plants, 2019, doi:10.3390/plants8060181_

Round 1
Reviewer 1 Report
Even if the authors answered most of my comments, I have to say that I was quite disappointed by reading the revision of this manuscript. Indeed, the authors claim that they made corrections that I didn’t find in the text. I am mentioning the extra paragraph in the “Material and method”, the letters for statistical significance and the “µ” in the last paragraph of the “Material and methods” part. Please integrate your answers to my comments in a future version of the manuscript.
I appreciate that the authors added qPCR quantification for numerous new genes in Figure 4 but this figure is now just unreadable. Moreover, the authors claim that all genes tested were upregulated but just looking at the figure it does not seem to be the case (i.e. genes n°33 and 44 starting from the left). So first the authors should provide a more readable figure and second they should be more precise in the interpretation of the results. Finally as the statistical analysis is still missing (for qPCR and Digital PCR), it is difficult to evaluate the gene expression variations over exposure time and therefore to discuss the results in an appropriate manner.
Regarding the discussion part, the authors made several changes. However, the results presented still do not allow to support the hypothesis/conclusions, leading therefore to a discussion that is too speculative.
The authors should also be careful when citing literature. For instance, lines 308-309, they are citing of the work of Nardi et al. on Arabidopsis thaliana but if I am correct this work has been done on maize !!!
Finally, there is still several spelling errors in the Discussion part (i.e. lines 308, 336, 340, 341, 354…). Please carefully edit this part for the English.
Author Response
Even if the authors answered most of my comments, I have to say that I was quite disappointed by reading the revision of this manuscript. Indeed, the authors claim that they made corrections that I didn’t find in the text. I am mentioning the extra paragraph in the “Material and method”, the letters for statistical significance and the “µ” in the last paragraph of the “Material and methods” part. Please integrate your answers to my comments in a future version of the manuscript.
Author’s response: All corrections are now well-highlighted. We added statistical significance in the Figure 3. We added ‘μl’ where missing.
I appreciate that the authors added qPCR quantification for numerous new genes in Figure 4 but this figure is now just unreadable. Moreover, the authors claim that all genes tested were upregulated but just looking at the figure it does not seem to be the case (i.e. genes n°33 and 44 starting from the left). So first the authors should provide a more readable figure and second they should be more precise in the interpretation of the results. Finally as the statistical analysis is still missing (for qPCR and Digital PCR), it is difficult to evaluate the gene expression variations over exposure time and therefore to discuss the results in an appropriate manner.
Author’s response: We improved the resolution of Figure 4. Two genes, AI0IYVQ and AIAA083, showed amplification problems for respectively one and two of the exposure times. The problem could be related to primer annealing specificity.
We added the following part to the results section:
The expression level of 53 genes, putatively involved in nutrient metabolism, was evaluated in untreated and treated plants. The ANOVA showed a significant effect of treatment (*P>0.05), time of exposition (*P>0.05) and gene (**P<0.01) as reported in table 4.
Table 4. Analysis of variance (ANOVA) showing the effect of different treatments, time of exposition and genes (*P>0.05; **P<0.01, factorial ANOVA test) on the expression of 53 sugar beet genes putatively involved in nutrient metabolism.
Effect | df | SS | MS | F | P |
Treatment | 1 | 1421 | 1421 | 115.2 | ** |
Exposure | 2 | 2723 | 1362 | 231.3 | ** |
Gene | 52 | 4523 | 87 | 6.9 | * |
We improved the sentence as follow:
Wells with yellow dots indicated ROX signal (passive reference) while wells with red dots indicated VIC signal, correlated with the presence and quantity of targeted genes. QuantStudio 3D Digital PCR Analysis Suite Cloud Software calculated the absolute number of target and perform statistics, calculating confidence interval and precision. Precision is defined as spread of confidence interval around two sample concentrations at a given confidence interval. The overall precision of the chips analysed is 5%. Red dots could be described as number of dots or converted by the software into copies μl-1 of target.
Regarding the discussion part, the authors made several changes. However, the results presented still do not allow to support the hypothesis/conclusions, leading therefore to a discussion that is too speculative.
Author’s response:
We added the following part to the discussion:
This set of 53 genes has been chosen since they proved to change their expression in response to sulfate availability and microalgae treatments. Genes selected represents many metabolic functions that change according to the treatments. In order to better describe the metabolic changes, more focused gene panel should be created ad-hoc or by increasing the number of genes analysed. In this work, we choose to have a picture of the plants response, at a relatively low cost, but able to determine whether or not a biostimulant product works.
Similarly, to Barone et al. [21] the upregulation involves mainly the nutritional related genes since BLACKJAK increase root length and as a consequence nutrient uptake. In fact, BLACKJAK unregulated genes linked to sulfur metabolism and phosphate transporter as well as genes involved in cell organization.
The authors should also be careful when citing literature. For instance, lines 308-309, they are citing of the work of Nardi et al. on Arabidopsis thalianabut if I am correct this work has been done on maize !!!
Author’s response: This error was fixed.
Finally, there is still several spelling errors in the Discussion part (i.e. lines 308, 336, 340, 341, 354…). Please carefully edit this part for the English.
Author’s response: Several spelling errors in the discussion section were corrected.
Reviewer 2 Report
This manuscript was mainly about study the effect of leonardite-based biostimulant on sugar beet. The authors did onomic analysis and gene expression analysis as well as morphological traits evaluation. The experiment designs are reasonable and the results are well presented and I recommend this manuscript to be accepted after minor revision after grammar and spelling check.
Author Response
This manuscript was mainly about study the effect of leonardite-based biostimulant on sugar beet. The authors did onomic analysis and gene expression analysis as well as morphological traits evaluation. The experiment designs are reasonable and the results are well presented and I recommend this manuscript to be accepted after minor revision after grammar and spelling check.
Author’s response: Several spelling errors in the manuscript were corrected.
Reviewer 3 Report
Authors studied essential information to understanding the effect of HS from leonardite on plant growth hormonal metabolism, although the molecular and physiological basis for these complicated regulatory mechanisms. Although the overall interest and visibility of this work, some aspects should still be considered to improve the quality and objectiveness of this work.
· Please provide Primer sequence details.
· Background of the study should be made to very clear.
· Provide more details of introduction and review of the work.
· Materials and methods section provide detail information.
· Suggest changing the all Figures to make it clearer.
· Overall, this manuscript needs more discussion about experimental results.
· Conclusion part needs to be improved.
Author Response
Authors studied essential information to understanding the effect of HS from leonardite on plant growth hormonal metabolism, although the molecular and physiological basis for these complicated regulatory mechanisms. Although the overall interest and visibility of this work, some aspects should still be considered to improve the quality and objectiveness of this work.
Please provide Primer sequence details.
Author’s response: We added these details (please see supplementary material).
Background of the study should be made to very clear.
Author’s response: We improved it as follow:
Since the biostimulants sector is growing exponentially, it is necessary to deepen their mechanism of action. Particularly, complex effects of HS activity on sugar beet are poorly understood and need rigorous evaluation. In this work, we evaluated the responses of root traits and the expression of the nutritional related gene panel to the application of a leonardite-based biostimulant in sugar beet grown in Hoagland’s solution under controlled conditions.
Provide more details of introduction and review of the work.
Author’s response: We add the following sentences:
In tomato, wheat and maize HS enhance lateral roots and improve seedling root growth (Tahir et al 2011, Jindo et al 2012), while in bean plants, rice and pepper HS improve tolerance under salt stress (Aydin et al 2012, Cimrin et al 2010, Garcia et al 2013).
Leonardite application have shown to improve Fe nutrition, N and P uptake, enhancing root growth and crop yield (Pertuit et al 2001). Moreover, Akinremi et al (2000) found that leonardite increased canola yield by supplying S directly and facilitating the uptake of other nutrients.
Moreover, at molecular level, the use of a selected gene panel allowed to study the difference in genetic expression of plants treated and untreated with the two microalgae extracts. The same gene panel has been successfully used to evaluate sugar beet responses to changes in sulfate availability (Stevanato et al 2018).
Materials and methods section provide detail information.
Author’s response: We added the following section:
Preliminary investigation
In order to choose the best BLACKJAK dose, two preliminary tests were conducted. The first test was performed by using the following BLACKJAK dilutions: 1:10, 1:100, 1:1,000 and 1:10,000. The parameter measured was the root length (after 72 h of treatment) in seedlings grown as described in the next paragraph. The highest value of root length was recorded by using BLACKJAK 1:1,000 (data not shown). To validate this result, the second test was performed in the same experimental condition in a short range (around 1:1,000) by using the following dilutions: 1:700, 1:1,000, 1:1,300 and 1:1,600. In this second experiment the following root parameters software were measured by WinRHIZO (as described in below paragraph): length, surface area, tips and forks (Data not shown). In this validation trial the best dilution was BLACKJAK 1:10,00, as well. In the controls, the BLACKJAK was replaced with water.
Suggest changing the all Figures to make it clearer.
Author’s response: We changed the figures.
Overall, this manuscript needs more discussion about experimental results.
Author’s response: The discussion section was improved and discussed according to the figures and tables showed in the results.
Conclusion part needs to be improved.
Author’s response: The conclusion section was improved according to the guidelines of the journal (“This section is not mandatory, but can be added to the manuscript if the discussion is unusually long or complex”).
Round 2
Reviewer 1 Report
Thanks to the authors for answering my comments. I do not have any additionnal to make.
This manuscript is a resubmission of an earlier submission. The following is a list of the peer review reports and author responses from that submission.
Round 1
Reviewer 1 Report
The manuscript is of a very good quality and the subject is interesting. I am upploading the entire text with very few corretions/suggestions.
Take a few minutes to comply.

Reviewer 2 Report
This manuscript was mainly about the effect of leonardite on the molecular and morphology change of sugar beet. The main results of this manuscript were the morphology results and transcript level detection of PIN, LOGL10 and GID1. However, such results are not enough for this manuscript to be published in an international journal.
Reviewer 3 Report
The manuscript entitled « Molecular and morphological changes induced by leonardite-based biostimulant in Beta vulgaris » aims at assessing the effect of a commercial product named BLACKJAK on the root development of sugar beet at its early stage of development. Additionally, the authors used qPCR and digital PCR to quantify the expression of 3 genes related to hormones transport/signaling in untreated and treated roots. Overall the authors conclude that the application of BLACKJAK improves efficiently the root development of the sugar beet at early stage of development.
The manuscript is globally well-written and easy to follow. Overall, I felt that this manuscript is preliminary and therefore is a good starting point for a wider study. Then I wondered if there is enough data to be published in this state. Nonetheless, I have several points that the authors should address to improve the overall quality of this work.
- The authors should better justify the use of the 1/1000 dilution of BLACKJAK. Is this concentration is commonly used in real field context?
- I am not a NMR specialist but I found weird that the total of the % provided in Table 2 is not equal to 100%. Could the authors give an explanation?
- The authors are not at all referring to Figure 1B in the text. Is this panel needed? If yes, please provide a description for it and the assignments for the different wavenumbers.
- Line 108: The authors should provide the data for the elemental composition of the Hoagland medium.
- Table 3. I found really strange that in treated roots, the quantity of all the element was at least 2-fold less than in untreated roots. Especially since the authors show that root growth is improved by the treatment. It looks to me that some dilution effect is going on (in the calculation maybe?) Are these results correct? If so, the authors could discuss this point in the text.
- Figures 3 and 4: Letters for statistical analysis are missing. Please provide them.
- Figures 5, 6 and 7: To what yellow and red colors stand for?
- Line 257: I disagree with the authors that their results are in accordance with the fact that cytokinins and auxins act as antagonist. Indeed, they only quantified the expression of one gene related to cytokinins biosynthesis and of one gene related to auxin transport and even if the expression one of them is decreasing while the other one is increasing, it is absolutely not enough to make this statement. The authors should quantify the level of the different hormones, along with the expression data, to strengthen their conclusion. The same comment is true when authors are discussing the root phenotype along with the expression data. Knowing that the root and lateral root development are complex mechanisms involving multiple hormonal crosstalk temporally and spatially, I found difficult to make correlation between both with this king of results.
- Lines 387-393: several “µ” are missing for the units